# The Novel Variant NP_00454563.2 (*p.Glu259Glyfs*77*) in Gene *PKP2* Associated with Arrhythmogenic Cardiomyopathy in 8 Families from Malaga, Spain

**DOI:** 10.3390/genes14071468

**Published:** 2023-07-19

**Authors:** Ainhoa Robles-Mezcua, Amalio Ruíz-Salas, Carmen Medina-Palomo, María Robles-Mezcua, Arancha Díaz-Expósito, María Victoria Ortega-Jiménez, Juan Ramón Gimeno-Blanes, Manuel F. Jiménez-Navarro, José Manuel García-Pinilla

**Affiliations:** 1Heart Failure and Familial Cardiomyopathies Unit, Cardiology Department, University Hospital Virgen de la Victoria, Instituto de Investigación Biomédica de Málaga (IBIMA), 29010 Málaga, Spain; amalioruizsalas@gmail.com (A.R.-S.); c.medinapalomo@gmail.com (C.M.-P.); mariaroblesmezcua@gmail.com (M.R.-M.); aranchadiazexposito@gmail.com (A.D.-E.); mjimeneznavarro@gmail.com (M.F.J.-N.); marlucale41@gmail.com (J.M.G.-P.); 2Centro de Investigación Biomédica en Red en Enfermedades Cardiovasculares (CIBERCV), Instituto de Salud Carlos III, 28220 Madrid, Spain; juanramon.gimeno@gmail.com; 3Pathological Anatomy Service, IBIMA, 29590 Málaga, Spain; mariavi715@yahoo.es; 4Human Physiology, Human Histology and Physical and Sports Education Department, Universidad de Málaga, 29071 Málaga, Spain; 5Inherited Cardiac Disease Unit, Cardiology Department, University Hospital Virgen de la Arrixaca, El Palmar, 30120 Murcia, Spain

**Keywords:** arrhythmogenic cardiomyopathy, *PKP2*, genetic variant, arrhythmic events, heart failure

## Abstract

Introduction and objectives: Arrhythmogenic cardiomyopathy (ACM) is a hereditary heart disease defined by the progressive replacement of the ventricular myocardium with fibroadipose tissue, which can act as a substrate for arrhythmias, sudden death, or even give rise to heart failure (HF). Sudden death is frequently the first manifestation of the disease, particularly among young patients. The aim of this study is to describe a new pathogenic variant in the *PKP2 gene*. Methods: A descriptive observational study that included eight initially non-interrelated families with a diagnosis of ACM undergoing follow-up at our HF and Familial Cardiomyopathies Unit, who were carriers of the NM_004572.3:c.775_776insG; *p.(Glu259Glyfs*77)* variant in the *PKP2 gene*. The genetic testing employed next-generation sequencing for the index cases and the Sanger method for the targeted study with family members. We compiled personal and family histories, demographic and clinical characteristics, data from the additional tests at the time of diagnosis, and arrhythmic events at diagnosis and during follow-up. Results: We included 47 subjects, of whom 8 were index cases (17%). Among the evaluated family members, 16 (34%) were carriers of the genetic variant, 3 of whom also had a diagnosis of ACM. The majority were women (26 patients; 55.3%), with a mean age on diagnosis of 48.9 ± 18.6 years and a median follow-up of 39 [24–59] months. Worthy of note are the high incidences of arrhythmic events as the form of presentation and in follow-up (21.5% and 20.9%, respectively), and the onset of HF in 25% of the sample. The most frequent ventricular involvements were right (four patients, 16.7%) and biventricular (four patients, 16.7%); we found no statistical differences in any of the variables analysed. Conclusions: This variant is a pathogenic variant of gene *PKP2* that has not previously been described and is not present in the control groups associated with ACM. It has incomplete penetrance, a highly variable phenotypic expressivity, and was identified in eight families of our geographical area in Malaga (Andalusia, Spain), suggesting a founder effect in this area and describe the clinical and risk characteristics.

## 1. Introduction

Arrhythmogenic cardiomyopathy (ACM) is a heart disease with a genetic basis, defined by the progressive replacement of the ventricular myocardium with fibroadipose tissue, which can act as a substrate for arrhythmias and sudden death (SD), or even give rise to heart failure (HF). Estimates place the prevalence of ACM among the general population at 1:3000–5000 inhabitants. It generally presents between the ages of 20 and 40 years, and usually manifests in the form of palpitations, syncope, or presyncope, with SD often the first manifestation of the disease, particularly in the case of young patients [1,2].

Despite classically having been described as arrhythmogenic right ventricular dysplasia (ARVD), frequent biventricular or predominant left ventricle involvement have led to adoption of the term ACM [3]. Both the biventricular and the predominant left ventricular forms can be difficult to differentiate from dilated cardiomyopathy (DCM) [3,4].

Complex molecular mechanisms exist in the physiopathology of ACM that differ according to the genotype [5,6]. Classically, ACM has been considered a disease of autosomal dominant inheritance with reduced penetrance and variable clinical expressivity depending on age, caused by mutation in the genes that encode desmosomal proteins [7,8]. There are five genes that encode these proteins and are involved in the pathogenesis of ACM: plakophilin-2 (*PKP2*), desmoglein-2 (*DSG2*), desmoplakin (*DSP*), desmocollin-2 (*DSC2*), and junction plakoglobin (*JUP*). The literature also describes certain, much less frequent, forms with autosomal recessive transmission, which are usually syndromic [9]. Approximately 40–60% of cases with a definitive ACM diagnosis show at least one genetic variant that could explain the disease [10], for the most part in a desmosomal gene, although in a low percentage of cases it is possible to identify causal variants in non-desmosomal genes (*DES*, *PLN*, *TMEM43*, *LMNA*, *TTN*, *SCN5A*, *CTNNA3*, *CDH2*, *TJP1*, *ANK2*, *TP63*) [11]. Recent years have seen numerous cases of ACM described as oligogenic or even multifactorial, with genomic and atmospheric factors playing a role in pathogenesis [10,11].

The clinical diagnosis of ACM is not an easy task, given that it shares characteristics with many other diseases such as DCM, myocarditis, sarcoidosis, and some variations of normality such as athletic heart syndrome [12]. Genetic diagnosis is also complex, due to the high predominance of variants that are missense-type or of uncertain significance in desmosomal genes, particularly in *PKP2* [13], which means that interpretation of the genetic study must be undertaken in conjunction with clinical data and a familial study to establish the presence or not of cosegregation. Thus, the 2010 revised diagnostic criteria for ACM included genetic criteria [14].

The aim of our study is to describe the clinical, genetic, and epidemiological characteristics of eight initially non-interrelated families who have a diagnosis of ACM associated with the *NM_004572.3:c.775_776insG; p.(Glu259Glyfs*77)* variant in the *PKP2 gene*. This variant has not previously been described, and we could establish its pathogenicity and a founder effect in our region in Malaga (Andalusia, Spain) that allows us to evaluate the genotype–phenotype correlation, and describe the clinical and risk characteristics of both index cases and family members in follow-up at our HF and Familial Cardiomyopathies Unit.

## 2. Methods

### 2.1. Study Design and Population

This is a descriptive observational study that included 8 families with a diagnosis of ACM related with the genetic variant *NM_004572.3:c.775_776insG; p.(Glu259Glyfs*77)* in gene *PKP2* who underwent follow-up at our HF and Familial Cardiomyopathies Unit. We evaluated 8 probands, or index cases, who had undergone genetic testing as part of the habitual study of their heart disease. We also included 39 family members of these index cases, performing cascade testing. We broadened the initial evaluation to include further tests in cases of suspicion or indications of disease. The informed consent of each subject was obtained, and the study protocol met the ethical criteria of the 1975 Declaration of Helsinki.

The definitive diagnosis of ACM followed the 2010 modified Task Force criteria [14]. We compiled personal and family histories, demographic and clinical characteristics, additional tests at the time of diagnosis, and arrhythmic events both at diagnosis and during follow-up. 

### 2.2. Genetic Analysis 

The study of the index cases of each family was performed by Next Generation Sequencing (NGS) using a library that included 121 genes related to ACM and DCM as a part of the habitual clinical evaluation. The genes included in this test were selected based on clinical criteria, taking into account their association with a given phenotype (among them, we highlight *ACTA1*, *BAG3*, *BRAF*, *DES*, *DMD*, *DSC2*, *DSG2*, *DSP*, *EMD*, *FHOD3*, *FLNC*, *GLA*, *JUP*, *KCNJ2*, *LMNA*, *MYBPC3*, *MYH6*, *MYH7*, *PKP2*, *PLN*, *RBM20*, *RYR2*, *SCN5A*, *TMEM43*, *TTN*, and *TTR*). In addition, the method is combined with the Sanger technique for those regions with suboptimal coverage and/or quality levels, which are re-sequenced using this technique. By this method it is possible to identify point substitutions and small insertions/deletions of up to 20 nucleotides with a sensitivity and specificity of the method higher than 99%.

For the study of the *PKP2* variant in family members, we used the Sanger method, sequencing only the variant under study. We obtained the blood samples used for the genetic testing after receiving signed informed consent from each subject. The pathogenicity of the genetic variants was classified according to the American College of Medical Genetics and Genomics [15].

### 2.3. Clinical Evaluation and Follow-Up

For all index cases identified, we exhaustively researched the family history and prepared a family tree covering at least 3 generations, which allowed us to carry out the cascade familial screening of all accessible family members. All subjects received a clinical evaluation that included a personal and family history, a physical examination, a 12-lead ECG, and a transthoracic echocardiogram. The 24-h Holter monitoring and cardiac magnetic resonance imaging (CMRI) were only undertaken for the index cases and for family members who were affected or had suspected incipient disease, provided no contraindications existed for the study (claustrophobia or wearers of implantable cardioverter–defibrillators—ICDs—or other devices). The clinical information was retrospectively recorded for patients who had a previous diagnosis, and prospectively for new cases or evaluated family members. 

Thus, we identified, treated, and established follow-up for all carrier subjects of the *NM_004572.3:c.775_776insG; p.(Glu259Glyfs*77)* variant, in accordance with the latest evidence available for ACM and DCM [16,17].

During follow-up, we collected information on arrhythmic events, including sustained ventricular tachycardia (SVT) and non-sustained ventricular tachycardia (NSVT), appropriate ICD therapies (with anti-tachycardia therapy or shock) and aborted SD. We also considered the development of heart failure (HF), the need for heart transplant, and death. 

### 2.4. Statistical Analysis

We used SPSS (version 22) to analyse the data. Quantitative variables are expressed as a mean ± standard deviation, or as a median and interquartile range according to whether they followed a normal distribution or not. Categorical variables are expressed as an absolute value and a percentage. For comparison of continuous variables, we used Student’s *t*-test or the Mann–Whitney U test for normal or non-normal distributions, respectively. Categorical variables were compared by means of contingency tables and the x2 test or Fisher’s exact test. Two-tailed values of *p* < 0.05 were considered statistically significant.

## 3. Results

### 3.1. Baseline Characteristics of the Sample

Of the 47 subjects studied, 8 were index cases or probands (17% of the sample) who met the diagnostic criteria for ACM and were carriers of the NM_004572.3:c.775_776insG; *p.(Glu259Glyfs*77)* variant in gene *PKP2*. The 39 family members evaluated underwent a targeted genetic study for the variant and 16 (41%) were found to be carriers, 3 of whom (19%) were also affected by heart disease: one subject with incipient involvement, whose data it was not possible to complete due to exitus related to gastric cancer; another patient with a phenotype overlapped with hypertrophic cardiomyopathy (HCM) and with a complex genotype, who was additionally a carrier of pathogenic variants in sarcomeric genes; and another patient with incipient left-dominant ACM. 

Most of the subjects evaluated were women (26 patients: 55.3%), with a mean age on diagnosis of 48.9 ± 18.6 years. The left ventricular ejection fraction (LVEF) was 64% [60–65] and the follow-up period was 39 [24–59] months. Table 1 describes the baseline characteristics of the sample, and Table 1B illustrates the main clinical characteristics of the subjects who are carriers of the genetic variants and were diagnosed with ACM (11 subjects, 17.2% of the sample). Among this group, in most cases, presentation of the disease was for arrhythmic events: four (36.3%) patients with VT (both left and right bundle branch block morphology depending on myocardial involvement), one (9.1%) with aborted SD, and another (9.1%) for palpitations caused by high-density ventricular ectopic beats. In addition, three (27.2%) of the diagnoses were made for symptoms of HF and another two (18.2%) on conducting the familial studies. Right ventricular involvement was the most common (four patients, 36.3%), alongside biventricular involvement (four patients, 36.3%), and only three patients (27.2%) had exclusive left involvement. None of the subjects with right involvement had a right ventricular dysfunction that was greater than mild, and right involvement was considered for segmental hypokinesia. 

We also detected variants in genes other than *PKP2* in nine patients. In the first five families (Figure 1) no other variants in other genes were identified. In family 6 (Figure 2) we identified a variant in the *DSG2* gene that has been described to be associated with the development of ACM in homozygosis or compound heterozygosis, it being highly probable that it contributes to the development of ACM in the presence of another pathogenic variant as is our case in *PKP2* and more specifically in the index case of this family. In addition, we found another variant in the *DSP gene* considered to be of uncertain pathogenicity and a more extensive cosegregation study is necessary. In family 7 (Figure 2) two variants in the gene that encodes myosin binding protein C3 (*MYBPC3*) were found to be associated with the development of HCM, unrelated to DCM and/or ACM. In this family, the HCM phenotype is only evident in patients carrying these variants in *MYBPC3*. In Family 8 (Figure 2) we found variants of unknown pathogenicity in the inward-rectifier potassium channel gene (*KCNJ2*) and titin gene (*TTN*), and another variant in the ryanodine receptor 2 gene (*RYR2*) which, although also considered of unknown pathogenicity, could contribute to the ventricular arrhythmia phenotype in the presence of other disease-causing variants such as the *PKP2* variant. Table 2 describes the genetic and clinical characteristics of the subjects diagnosed with ACM. 

It should be highlighted that, of the 47 patients studied, only 10 (21.3%) underwent CMRI and only one person had positive late gadolinium enhancement. In most cases, the reason that CMRI was not possible was claustrophobia. 

### 3.2. Follow-Up

During the mean follow-up period of 48 [27–59] months for the patients who were carriers of the described genetic variant, five patients (20.9%) suffered arrhythmic events, four (16.7%) as SVT and one (4.2%) as NSVT. Only two patients (8.3%) underwent an ablation procedure for VT; and eight patients (33.3%) had an ICD implant, four (16.7%) as primary prevention and the other four (16.7%) as secondary prevention (two subjects with right involvement and the other two with biventricular involvement). Regarding ICD therapies, two patients (8.3%) received appropriate therapies for SVT during follow-up. 

The onset of HF occurred in six patients (25%), one of whom eventually required a heart transplant with a good subsequent progression. 

We did not find any statistically significant differences between the sexes regarding events during follow-up (Table 1B), neither were there any differences in LVEF between the sexes (63% [40–65] among men and 64% [60–65] in women, *p* = 0.68), although we did find a statistical significance for age on diagnosis, which was lower for women (48.4 ± 17.3 years, *p =* 0.03). 

During follow-up, four patients died, one of them due to HF and three others not directly related to ACM: in two cases it was due to oncological processes (gastric cancer and leukaemia) and the third was multifactorial in an elderly patient. 

## 4. Discussion 

*PKP2* encodes plakophilin-2, the protein that localises to desmosomes and nuclei, and is involved in regulation of cellular signalling and in cell–cell junctions through intermediate filaments in the cytoskeleton. It is expressed in various tissues, including the myocardium, and high levels of this protein are detected in the RV. The intention of this study is to describe our carrier population of the *NM_004572.3:c.775_776insG*; *p.(Glu259Glyfs*77)* variant in *PKP2*, which has been identified among eight families of our geographical area in Malaga (Andalusia, Spain) who were initially non-interrelated but for whom we could establish a founder effect. This variant has not previously been described or reported in the databases used as population control (Exome Sequencing Project, 1000 Genomes Project, ExAC, etc.). It is also a radical variant provoking truncation in the protein, which causes functional deficit and/or premature degradation. Radical variants (nonsense, frameshift, and splicing) are considered clearly pathogenic [15] as they cause a premature stop codon and are most frequently associated with the pathogenesis of ACM, although deletions and duplications have also been identified as the cause of the phenotype. 

Pathogenic variants in *PKP2* are one of the most frequent causes of arrhythmogenic cardiomyopathy and research suggests that such variants are responsible for between 30 and 50% of all ACM cases, although there is disparity between the different studies, probably due to geographical distribution [18,19]. Some series have found mutations of this gene in up to 70% of patients with a family history, above all in the Netherlands [18] with proportions that are higher than other registries for other parts of the world [20]. 

As in our sample, the incidence of desmosomal mutations described in the different published series varies between 30 and 60% [10,21,22]. The work of Ruiz-Salas et al. [23] indicates a higher incidence, most likely due to the selection of the sample, all of whom had received a definitive diagnosis of ACM. Likewise, the Medeiros-Domingo et al. [24] paper detected a higher percentage of mutation in desmosomal genes among patients with definitive RVACM compared to borderline or possible RVACM, who additionally had other variants in genes associated with ACM; the supposition was that this could consist of overlapped ACM and DCM or phenocopies.

Our research reports a previously undescribed genetic variant suggesting a founder effect in our geographic area. The description of new mutations in ACM registries is not always specified and as many as 25–70% of all of the pathogenic variants reported in the main series have not been previously described [25,26]. As we have already mentioned, there is wide geographic variability in the distribution of *PKP2* mutations [18] with multiple founder mutations described in areas such as the Netherlands [27]. These data support the suspicion that the genetic spectrum of ACM may have singularities in each geographical zone, and that large-scale registries and studies are necessary to establish the genotype–phenotype relationship.

Our series seems to suggest incomplete penetrance, with most of the affected carriers over the age of 55 years. Although this variant is sufficient to cause presentation of the disease, our cases with a phenotype at an earlier age are usually women or cases in which a second impact exists, such as the presence of another pathogenic genetic variant or participation in sports activities. Thus, our data concur with others described in the literature, with a variable expressivity of the disease among patients with *PKP2* mutations, with some very severe forms and others that are milder within the same family, with incomplete penetrance, depending on age, but differing in terms of sex, as these earlier and more severe cases have normally been described in males [28].

In most cases of ACM associated with *PKP2* mutations, right involvement predominates, and predominantly left forms, re rarer, although some reports suggest that the possibility of left involvement may be as high as 60% of carriers of pathogenic mutations in *PKP2* [29]. The previously cited Ruiz-Salas et al. study [23] found an association between *PKP2* mutations and cases of exclusively right ventricle involvement, whereas mutations in DSP correlated with predominantly left ventricle involvement. Different series, such as that of Bhonsale et al. [30], have also observed the same results; likewise, our series found the most frequent ventricular involvement to be either right or biventricular (16.7% in both cases). 

The phenotype of these patients is characterised by presentation with ventricular arrhythmia, generally from youth [31], and several studies report an association between the presence of these mutations with a higher likelihood of arrhythmic events and presentation of the disease at an earlier age in comparison to patients with a negative genetic study, as occurred in the den Haan et al. series [24,25], which detected more VT events and an earlier age of diagnosis for carrier patients (73% versus 44%, and 33 versus 41 years). In our research, we found a high rate of arrhythmic events as the form of presentation (25.1%) and in follow-up (20.9%), in addition to appropriate ICD therapies and the need for ablation for VT (8.3% respectively), taking into account that this is a very select group, as occurred in the Ruiz-Salas et al. series [22], where the most frequent presentation of the disease was with arrhythmic events; 86% of the sample presented an arrhythmic episode throughout their lives. However, Bhonsale et al. [29], who published the largest series of RVACM genetic mutation carrier patients, did not find a worse arrhythmic profile or worse prognosis than for non-carriers, which could be attributable to the different profile of their sample with only 60% definitive RVACM. Among our series, the high rate of HF during follow-up (25%) is notable, despite the fact that this phenotype has more frequently been described in cases of ACM with mutations in genes DSG2 and DSP, with predominantly left involvement [30].

One of the challenges currently facing cardiology remains the prognosis and risk stratification for sudden death in ACM [16], as there is a considerable lack of coherence between the factors associated with a greater risk of SD and/or ventricular arrhythmias in the studies and series of published cases. What has been generally reported is a lower incidence of ventricular arrhythmia among family members diagnosed during the familial screening in comparison with the index cases, as also occurs in our series, although recent studies appear to cast doubt on this [32]. 

Our research also seeks to highlight the importance of genetics in the study of cardiomyopathies and in particular of ACM, since not only does it allow us to confirm diagnosis, and indeed is one of the diagnostic criteria of the Task Force [14], but also to perform a familial screening for early detection of the disease and to establish recommendations for physical activities and work, thus avoiding unnecessary diagnostic tests or follow-up among patients with negative genetic testing results. In many cases, it also represents a guide for prognosis, with higher or lower risk of arrhythmias or of HF in some of the genetic mutations identified, as we have already indicated [30]. Nevertheless, we should not forget that genetic testing requires suitable interpretation in conjunction with clinical data by specialist personnel, since recent awareness of hereditary diseases and the new sequencing technology available mean that genetic testing is increasingly requested, and consequently the number of described variants associated with ACM is higher. 

### Limitations

This is a descriptive single-centre study, with a small number of patients and no control group, which does not allow extrapolation of the data to all patients who are carriers of *PKP2* variants. Moreover, as most of the data used are retrospective, confounders exist that may affect the results. 

Another limitation has been that some family members rejected a clinical or genetic study and, for the cases that underwent the study, the request for imaging tests indicated “familial screening for ACM” which may have introduced bias. 

In our study, we also refer to penetrance, but many of the subjects diagnosed were asymptomatic and diagnosis was by means of familial screening. 


**What is known?**
-ACM is a heart disease defined by the progressive replacement of the right ventricle myocardium with fibroadipose tissue, and may cause arrhythmias, sudden death, and heart failure. Currently, 40–60% of patients have at least one genetic mutation related with the disease.



**What is new?**
-This study presents a *PKP2* variant not previously described or present in the control population, and establishes a probable founder effect in our region. -This variant has incomplete penetrance and a highly variable phenotypic expressivity, and is strongly influenced by age, sex, and the presence of other genetic or environmental factors such as sport, characteristics that the majority of desmosomal gene mutations share.


## 5. Conclusions

Our study describes the *p.Glu259Glyfs*77* variant in the *PKP2 gene*, previously undescribed and not present among the control groups, which we identified in eight families of our geographical area in Malaga (Andalusia, Spain) who were initially non-interrelated, suggesting a founder effect in our region. This allowed us to evaluate the genotype–phenotype correlation and describe the clinical and risk characteristics of both the index cases and the family members under follow-up in our HF and ICD Unit. 

This variation is a pathogenic mutation associated with ACM; it has incomplete penetrance until advanced ages and with some male sex-dependent tendency; phenotypic expressivity is extremely variable and heavily influenced by the presence of other genetic or environmental factors, characteristics shared by most mutations in desmosomal genes.

## Figures and Tables

**Figure 1 genes-14-01468-f001:**
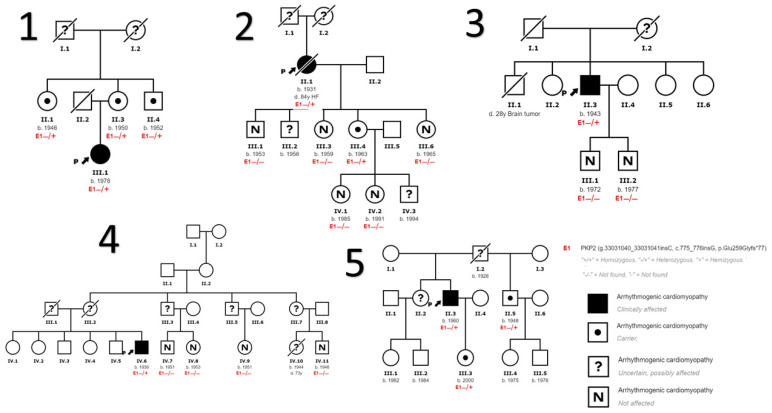
Family trees of families carrying only the *PKP2* variant. The index case is the subject marked with the arrow in each family.

**Figure 2 genes-14-01468-f002:**
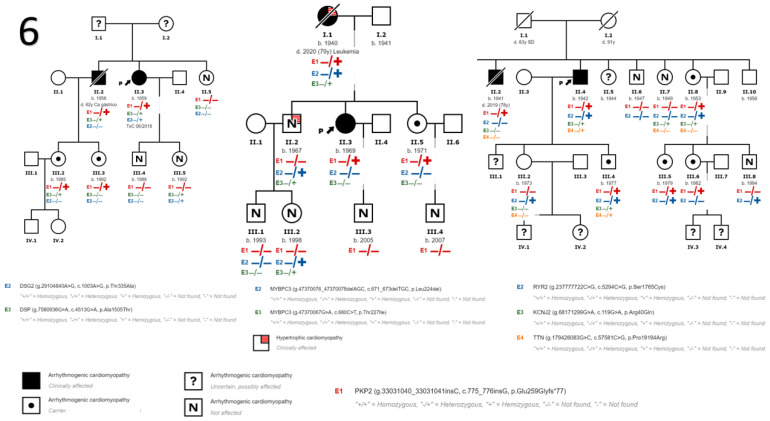
Family trees of families that in addition to the variant in *PKP2* have other variants in other genes. The index case is marked with an arrow in each family.

**Table 1 genes-14-01468-t001:** Baseline and follow-up characteristics.

A.	Carriers	Non-Carriers	*p*	Total
Sex			0.38	
Women	15 (31.9%)	11 (23.4%)		26 (55.3%)
Men	9 (19.1%)	12 (25.5%)		21 (44.7%)
Age (years)	54 ± 16.7	43.6 ± 19.3	0.06	48.9 ± 18.6
Follow-up time (months)	48 [27–59]	27 [23–59]	0.13	39 [24–59]
LVEF (%)	63 [45–65]	64 [62–65]	0.43	64 [60–65]
**B.**	**Men**	**Women**	** *p* **	**Total**
Diagnostic ACM/DCM	6 (54.5%)	5 (45.5%)	0.50	11 (100%)
Probands	4 (50%)	4 (50%)		8 (72.7%)
Family members	2 (66.7%)	1 (33.3%)		3 (27.2%)
Age at diagnosis (years)	63.2 ± 11.2	48.4 ± 17.3	0.03	54 ± 16.7
Form of presentation			0.26	
Family screening	2 (18.2%)	0 (0%)		2 (18.2%)
Dyspnoea	1 (9.1%)	2 (18.2%)		3 (27.2%)
Palpitations	0 (0%)	1 (9.1%)		1 (9.1%)
Arrhythmic events	3 (27.2%)	1 (9.1%)		4 (36.3%)
Sudden death	0 (0%)	1 (9.1%)		1 (9.1%)
Involvement			0.32	
Right	3 (27.2%)	1 (9.1%)		4 (36.3%)
Left	2 (18.2%)	1 (9.1%)		3 (27.2%)
Biventricular	1 (9.1%)	3 (27.1%)		4 (36.3%)
ECG			0.47	
RBBB	3 (27.2%)	3 (27.2%)		6 (54.4%)
T negative wave	1 (9.1%)	2 (18.2%)		3 (27.2%)
Epsilon wave	1 (9.1%)	0 (0%)		1 (9.1%)
LVEF (%)	63 [40–65]	64 [60–65]	0.68	63 [45–65]
Follow-up time (months)	59 [33–75]	42 [24–58]	0.07	48 [27–59]
ICD implant			0.25	
Primary prevention	1 (9.1%)	3 (27.2%)		4 (36.3%)
Secondary prevention	2 (18.2%)	2 (18.2%)		4 (36.3%)
Events in the follow-up			0.45	
SVT	2 (18.2%)	2 (18.2%)		4 (36.3%)
NSVT	0 (0%)	1 (9.1%)		1 (9.1%)
VT ablation	1 (9.1%)	1 (9.1%)		2 (27.2%)
ICD therapies	1 (9.1%)	1 (9.1%)		2 (27.2%)
HF	3 (27.2%)	3 (27.2%)		6 (54.5%)
Heart transplantation	0 (0%)	1 (9.1%)		1 (9.1%)

(**A**) Baseline characteristics of the sample. (**B**) Clinical characteristics of subjects carrying the variant *NM_004572.3:c.775_776insG; p.(Glu259Glyfs*77)* in the *PKP2 gene* and diagnosed with ACM. Values expressed as *n (%)* or *median [interquartile range]*. ECG: electrocardiogram; ICD: implantable cardioverter–defibrillator; LVEF: left ventricular ejection fraction; HF: heart failure; NSVT: non-sustained ventricular tachycardia; RBBB: right bundle branch block; SVT: sustained ventricular tachycardia; VT: ventricular tachycardia.

**Table 2 genes-14-01468-t002:** Clinical and genetic characteristics of patients diagnosed with ACM.

Family/ Patient	Sex	Age 1st Visit (Years)	Index Case	Other Genetic Variants	Presentation	Involvement	LVEF (%)	ICD	Arrhythmias under Follow-Up	HF	Death
1/III.1	Woman	34	Yes	No	Palpitations (VPB)	Right	65	Primary Prevention	SVT with focus ablation	No	No
2/II.1	Woman	76	Yes	No	VT	Biventricular	38	Secondary Prevention	SVT	Yes	Yes (HF)
3/II.3	Man	69	Yes	No	VT	Right	65	Secondary Prevention	SVT	No	No
4/IV.6	Man	73	Yes	No	VT	Right	65	No	No	No	No
5/II.3	Man	54	Yes	No	VT	Right	65	Secondary Prevention	SVT with focus ablation	No	No
6/II.3	Woman	56	Yes	DSP p.Ala1505Thr (+?) DSG2 p.Thr335Al (+++)	Dyspnea	Left	30	Primary Prevention	NSVT	Yes (CT)	No
6/II.2	Man	63	No	DSP p.Ala1505Thr (+?)	Family Study	Left	40	No	No	Yes	Yes (gastric cancer)
7/II.3	Woman	48	Yes	No	SD	Biventricular	65	Secondary Prevention	No	No	No
7/I.1	Woman	65	No	MYBPC3 p.Leu224del (++) * MYBPC3 p.Thr227Ile (++) *	Dyspnea	Biventricular	38	Primary Prevention	No	Yes	Yes (leukaemia)
8/II.4	Man	63	Yes	RYR2 p.Ser1765Cys (+?) KCNJ2 p.Arg40Gln (-?) TTN p.Pro19194Arg (-?)	Dyspnea	Left	40	No	No	Yes	No
8/II.2	Man	74	No	RYR2 p.Ser1765Cys (+?) KCNJ2 p.Arg40Gln (-?) TTN p.Pro19194Arg (-?)	Family Study	Biventricular	30	Primary Prevention	No	Yes	Yes (older age)

The genetic and clinical characteristics of the patients carrying the variant under study and diagnosed with ACM are shown. The 8 probands and the 3 patients diagnosed at the time of the family study are included. CT: cardiac transplant; HF: heart failure; ICD: automatic implantable defibrillator; *DSG2*: desmoglein-2; *DSP*, desmoplakin; LVEF, left ventricular ejection fraction; *KCNJ2*: potassium inward rectifier channel; SD: sudden death; *MYBPC3*: cardiac myosin-binding protein C; *TTN*: titin; VPB: ventricular premature beats; VT: ventricular tachycardia, NSVT: non-sustained ventricular tachycardia; SVT: sustained ventricular tachycardia; *RYR2*: ryanodine receptor; (-?) possibly NOT associated with the disease; (+?) possibly associated with disease; (++) very possibly associated with disease; (+++) pathogenic or associated with disease. * Genetic variants related to HCM, diagnosis also made in this family.

## Data Availability

Data available on request due to restrictions (privacy and ethical). The data presented in this study are available on request from the corresponding author.

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
