# Peer review of "The Novel Variant NP_00454563.2 (p.Glu259Glyfs*77) in Gene PKP2 Associated with Arrhythmogenic Cardiomyopathy in 8 Families from Malaga, Spain"

_genes, 2023, doi:10.3390/genes14071468_

Round 1
Reviewer 1 Report
In the submitted manuscript, the authors present the results of genetic studies of patients with ACM in the region of Malaga. In as many as 8 probands, they identified a novel PKP2 variant. This is a surprising find, raises the suspicion of a founder effect and is worth presenting.
However, the method of presentation is not transparent, sometimes it pays attention to insignificant aspects and omit important ones.
My concerns:
1. Typically, in order to demonstrate the founder effect, it is necessary to perform a haplotype analysis, which may allow to determine the degree of potential relatedness and 'the age' of the founder mutation - the authors do not mention such an analysis.
2. I believe that analyzing the phenotype of relatives who do not carry the variant does not provide significant information. Especially that people without clear indications for screening were also examined (e.g. 5 cousins from the Family 4). In my opinion, the phenotype of noncarriers can be crucial primarily when assessing the potential effect of concomitant variants (which they inherited).
3. The revised criteria allow us to establish definite, borderline and possible ARVC diagnosis. I don't think the categories used to describe the phenotype in Figure 1 are more legible.
3. The Gen + / Phen - group is much more interesting than noncarrier group. All variant carriers have actually "possible ARVC" (1 major criterion) and they don't need much to diagnose borderline or definitive disease. Why didn't you perform CMR and Holter monitoring in them to assess them comprehensively. I notice that there are many subjects at the age of 50+, 60+ or even 70+ among the variant carriers (this is worth highlighting as evidence of incomplete penetration) and potential findings in this group would be difficult to unambiguously classify as the result of ACM or other diseases. However, at least younger patients should be investigated more closely.
4. In ARVC, ECG abnormalities, e.g. negative T-waves in precordial leads, often precede changes in imaging modalities. In your cohort negative T waves were present in only 3/11 affected subjects and I also assume 13/16 variant (+) relatives had no diagnostic ECG anomalies. Would you comment on this?
5. The percentages on page 4 refer in my opinion to wrong groups, e.g.
line 150 - 16 carriers of 39 relatives - 41%
line 151 - 3 affected of 16 relatives with carrier status - 19%
6.The concept of simultaneously recognizing ACM and DCM in families 6 and 8 is not clear. Do you recognize them as separate entities and classify ACM as left, right or biventricular at the same time?
If you recognize DCM in families 6 and 8, how was the diagnosis of ACM with isolated LV involvement made?
7. Many of the terms used need to be clarified, e.g. RV, LV, BiV affectation (involvement?), "high-density VEBs", "arrhythmic events"
Author Response
In the submitted manuscript, the authors present the results of genetic studies of patients with ACM in the region of Malaga. In as many as 8 probands, they identified a novel PKP2 variant. This is a surprising find, raises the suspicion of a founder effect and is worth presenting.
However, the method of presentation is not transparent, sometimes it pays attention to insignificant aspects and omit important ones.
My concerns:
- Typically, in order to demonstrate the founder effect, it is necessary to perform a haplotype analysis, which may allow to determine the degree of potential relatedness and 'the age' of the founder mutation - the authors do not mention such an analysis.
It has not been possible to do so because many of the studies are old. We are conducting further research in this direction to complete this initial study.
- I believe that analyzing the phenotype of relatives who do not carry the variant does not provide significant information. Especially that people without clear indications for screening were also examined (e.g. 5 cousins from the Family 4). In my opinion, the phenotype of noncarriers can be crucial primarily when assessing the potential effect of concomitant variants (which they inherited).
I don't quite understand this comment. I am very sorry for this.
- The revised criteria allow us to establish definite, borderline and possible ARVC diagnosis. I don't think the categories used to describe the phenotype in Figure 1 are more legible.
The categories in Figure 1 are based only on genetics and the probability of developing the disease according to the age of the subjects.
- The Gen + / Phen - group is much more interesting than noncarrier group. All variant carriers have actually "possible ARVC" (1 major criterion) and they don't need much to diagnose borderline or definitive disease. Why didn't you perform CMR and Holter monitoring in them to assess them comprehensively. I notice that there are many subjects at the age of 50+, 60+ or even 70+ among the variant carriers (this is worth highlighting as evidence of incomplete penetration) and potential findings in this group would be difficult to unambiguously classify as the result of ACM or other diseases. However, at least younger patients should be investigated more closely.
Totally agree with this comment. That is why we continue to closely monitor all carriers, with imaging and holter tests. Those who do not undergo MRI are usually due to claustrophobia.
- In ARVC, ECG abnormalities, e.g. negative T-waves in precordial leads, often precede changes in imaging modalities. In your cohort negative T waves were present in only 3/11 affected subjects and I also assume 13/16 variant (+) relatives had no diagnostic ECG anomalies. Would you comment on this?
Annotated in the text.
- The percentages on page 4 refer in my opinion to wrong groups, e.g.
line 150 - 16 carriers of 39 relatives - 41%
line 151 - 3 affected of 16 relatives with carrier status - 19%
Annotated in the text.
6.The concept of simultaneously recognizing ACM and DCM in families 6 and 8 is not clear. Do you recognize them as separate entities and classify ACM as left, right or biventricular at the same time?
If you recognize DCM in families 6 and 8, how was the diagnosis of ACM with isolated LV involvement made?
We do not recognize them as different entities, but rather as a greater expression of arrhythmic events or ventricular dysfunction.
- Many of the terms used need to be clarified, e.g. RV, LV, BiV affectation (involvement?), "high-density VEBs", "arrhythmic events"
Annotated in the text.

Reviewer 2 Report
The authors reported on a new, pathogenic founder variant in gene PKP2 and phenotype consistent with ACM in 47 subjects from 8 families. Although the study findings have limited applications, the report is new and the clinical characterization of the cohort is almost exhaustive. There are some additional information that could be provided.
- Did any of the patients present with "hot-phase" ACM? I.e. troponin peaks with chest pain (baseline or follow-up).
- Did any patient receive immunomodulatory treatment? A quite recent report suggested that immunosuppression (classically used in autoimmune myocarditis) was effective in a young patient carrying variant in PKP2 gene. If not, please briefly comment on diagnosis and workup of hot-phase ACM in the discussion.
- Can the authors comment of ventricular arrhythmia morphology in relation to the substrate (left-right-biventricular)? And/or in relation to myocardial inflammation vs fibrosis?
- It is a major limitation that only a small proportion of cases underwent cardiac MRI. Did any patient undergo characterization by histology? Any alternative imaging to cardiac MRI? CT scan/FDG-PET/electroanatomical mapping?
- Did the authors perform a research on possible gene modifiers and/or environmental factors accounting for heterogeneous phenotypes?
- Did any patient showed overlap with LVNC or arrhythmogenic variant of mitral valve prolapse?
-Please comment also on the role of "continuous arrhythmia monitoring" (i.e. ICD or loop recorders) in contrast to traditional discontinuous monitoing by Holter ECGs in the detection of arrhyhtmias.
- Presentation of data can be improved and effectively summarizes by using significant pictures. Please provide multiple figures to show: 1) echocardiogram and MRI findings; 2) baseline ECG and arrhythmias; 3) any other remarkable findings from diagnostic workup.
Thank you for sharing your interesting experience.
Nice case series, it may warrant publication. However, presentation can be improved by adding new data / discussing relevant related topics in a modern view.
Author Response
The authors reported on a new, pathogenic founder variant in gene PKP2 and phenotype consistent with ACM in 47 subjects from 8 families. Although the study findings have limited applications, the report is new and the clinical characterization of the cohort is almost exhaustive. There are some additional information that could be provided.
- Did any of the patients present with "hot-phase" ACM? I.e. troponin peaks with chest pain (baseline or follow-up).
- Did any patient receive immunomodulatory treatment? A quite recent report suggested that immunosuppression (classically used in autoimmune myocarditis) was effective in a young patient carrying variant in PKP2 gene. If not, please briefly comment on diagnosis and workup of hot-phase ACM in the discussion.
In this series we have not had any cases with these characteristics and no patient patient received immunomodulatory treatment.
- Can the authors comment of ventricular arrhythmia morphology in relation to the substrate (left-right-biventricular)? And/or in relation to myocardial inflammation vs fibrosis?
Annotated in the text.
- It is a major limitation that only a small proportion of cases underwent cardiac MRI. Did any patient undergo characterization by histology? Any alternative imaging to cardiac MRI? CT scan/FDG-PET/electroanatomical mapping?
In most of the patients the ICD was implanted before the MRI and in other occasions there have been other limitations such as clautrophobia of the patients. We are aware of this major limitation.
- Did the authors perform a research on possible gene modifiers and/or environmental factors accounting for heterogeneous phenotypes?
- Did any patient showed overlap with LVNC or arrhythmogenic variant of mitral valve prolapse?
These issues are being studied in other research we are conducting to complete this initial work.
-Please comment also on the role of "continuous arrhythmia monitoring" (i.e. ICD or loop recorders) in contrast to traditional discontinuous monitoing by Holter ECGs in the detection of arrhyhtmias.
Depending on the patient's profile and situation, we use one or the other.
- Presentation of data can be improved and effectively summarizes by using significant pictures. Please provide multiple figures to show: 1) echocardiogram and MRI findings; 2) baseline ECG and arrhythmias; 3) any other remarkable findings from diagnostic workup.
This is already expressed in Table 1
Thank you for sharing your interesting experience.

Round 2
Reviewer 1 Report
The authors made minor changes to the manuscript, however, they did not significantly address the major issues, such as the lack of evidence for a founder effect or the lack of comparison of Phen(+) and Phen(-) carriers (probably due to poor phenotyping of the Phen- group).
My general assessment remains the same, that although the material is interesting, it needs to be presented in a different way, and even better supplemented with additional research.
My reservations include:
1. the term founder variant should not be presented as documented;
2. comparison of Phen(+) and Phen(-) carriers is lacking
3. how LV, RV, BiV involvement was determined?
4. the use of term DCM is confusing; left-dominant ACM could be more appropriate
5. the terms "not likely to manifest disease" "could later manifest disease" are imprecise, unclear andshould be avoided
Author Response
The authors made minor changes to the manuscript, however, they did not significantly address the major issues, such as the lack of evidence for a founder effect or the lack of comparison of Phen(+) and Phen(-) carriers (probably due to poor phenotyping of the Phen- group).
My general assessment remains the same, that although the material is interesting, it needs to be presented in a different way, and even better supplemented with additional research.
My reservations include:
We have made the suggested changes.
- the term founder variant should not be presented as documented;
We indicated that the variant could have a founding effect, but without affirming it.
- comparison of Phen(+) and Phen(-) carriers is lacking
We have not been able to make this comparison because we have few data from the Phen(-) carriers.
- how LV, RV, BiV involvement was determined?
Ventricular involvement was determined according to imaging tests (echocardiogram and MRI) as indicated in the text.
- the use of term DCM is confusing; left-dominant ACM could be more appropriate
We have removed the term DCM from the text and images.
- the terms "not likely to manifest disease" "could later manifest disease" are imprecise, unclear andshould be avoided
We have also removed these terms.
